# Exercise-Induced Electrocardiographic Changes in Healthy Young Males with Early Repolarization Pattern

**DOI:** 10.3390/diagnostics14100980

**Published:** 2024-05-08

**Authors:** Loránd Kocsis, Zsuzsanna Pap, Szabolcs Attila László, Hunor Gábor-Kelemen, István Adorján Szabó, Erhard Heidenhoffer, Attila Frigy

**Affiliations:** 1Department of Cardiology, Clinical County Hospital Mures, 540103 Targu Mures, Romania; lorand.kocsis@umfst.ro (L.K.); gabor_hunor@yahoo.com (H.G.-K.); sz.istvan.adorjan@gmail.com (I.A.S.); heiden.erhard@gmail.com (E.H.); attila.frigy@umfst.ro (A.F.); 2Department of Anatomy and Embryology, George Emil Palade University of Medicine, Pharmacy, Science and Technology of Targu Mures, 540103 Targu Mures, Romania; 3Department of Pneumology, Clinical County Hospital Mures, 540103 Targu Mures, Romania; szaby1994@gmail.com; 4Department of Internal Medicine IV, George Emil Palade University of Medicine, Pharmacy, Sciences and Technology of Targu Mures, 540103 Targu Mures, Romania

**Keywords:** early repolarization, exercise test, electrocardiogram, arrhythmia

## Abstract

*Background*: Exercise-induced modifications in ECG parameters among individuals with an early repolarization pattern (ERP) have not been evaluated in detail. We aimed to assess this phenomenon, with potential associations with arrhythmogenesis. *Methods*: Twenty-three young, healthy males with ERP (ERP+) participated in this study, alongside a control group, which consisted of nineteen healthy males without ERP (ERP−). ECGs at baseline, at peak exercise (Bruce protocol), and during the recovery phase were analyzed and compared between the two groups. *Results*: The treadmill test demonstrated strong cardiovascular fitness, with similar chronotropic and pressor responses in both groups. In the baseline ECGs, the QRS complex and the QT interval were shorter in the ERP+ group. During exercise, the P-wave duration was significantly longer and the QRS was narrower in the ERP+ group. In the recovery phase, there was a longer P wave and a narrower QRS in the ERP+ group. During the treadmill test, the J wave disappeared or did not meet the criteria required for ERP diagnosis. *Conclusions*: The slowed intra-atrial conduction found during exercise could be predictive of atrial arrhythmogenesis in the setting of ERP. The disappearing of J waves during exercise, due to increased sympathetic activity, has potential clinical significance.

## 1. Introduction

The early repolarization pattern (ERP) refers to the presence of a notch- or slur-like abnormality observable on an electrocardiogram (ECG) at the end of the QRS complex, also known as the J wave (Figure 1), which is traditionally considered to be a benign phenomenon. However, recent studies have confirmed an association between the ERP and life-threatening ventricular arrhythmias (early repolarization syndrome—ERS), particularly ventricular tachycardia (VT) and ventricular fibrillation (VF) [1,2].

The prevalence of ERP in the inferior and/or lateral leads ranges from 1% to 31% in the general population and is between 15% and 70% in individuals who have experienced VF [3,4,5]. It is more common in young people, males, and athletes [6,7].

The pathophysiological mechanism behind the generation of the J wave is not yet fully understood. According to the theory developed by Antzelevitch et al. [1], an increased notch appears in the first phase of the action potential in the epicardial myocardium compared to the endocardial layer. This abnormality has been explained, among other factors, by an increase in the transient outward potassium current, attributed to a genetic mutation. Due to the varied distribution of the transient outward current, an increase in the transmural voltage gradient occurs, which manifests as a J wave on the ECG. It has subsequently been proven that dysfunction in sodium, calcium, and ATP-dependent potassium channels can also lead to the appearance of a J wave [1]. According to Haissaguerre et al., there is evidence that structural abnormalities also contribute to the genesis of the J wave, which delay the conduction of impulses at the level of the epicardium, thus inducing the appearance of the J wave [8] (the “late depolarization” theory).

ERPs are associated with bradycardia, prolonged QRS duration, short QT intervals, and various other ECG abnormalities that have been proven to correlate with arrhythmias [9,10]. Notably, individuals with ERPs have a higher arrhythmia risk when the J-wave amplitude exceeds 2 mm or is located in the inferior or infero-lateral leads, especially if the J wave is followed by a horizontal or descending ST segment [3,7,11].

Physical activity is known to reduce cardiovascular risk and mortality, but it also significantly alters certain ECG parameters. Excessive exertion is linked to a higher incidence of sudden cardiac death during and post-exercise, making exercise-induced ECG changes a potentially valuable predictor of the arrhythmia risk [12,13,14,15,16].

Exercise-induced modifications in ECG parameters among individuals with an ERP have not been evaluated in detail. The objective of our study was to investigate the impact of exercise and the post-exercise recovery period on ECG parameters with potential predictive value for arrhythmic events in individuals with ERPs.

## 2. Materials and Methods

### 2.1. Study Population

Individuals aged between 18 and 28, with and without ERPs, identified from the ECG database of Mures County Clinical Hospital, Targu Mures, Romania, were invited to participate in our study. Twenty-three young (22.9 ± 1.6 years), healthy males who met the ERP criteria (ERP+ group) published by Macfarlane et al. [17] were included, alongside a control group of nineteen individuals (22.1 ± 1.9 years) without ERP (ERP− group).

Upon registration, each participant provided signed informed consent for participation, completed a comprehensive medical history questionnaire, and underwent a physical examination. All participants were free from known diseases, including cardiovascular conditions; exhibited normal findings during physical examinations; and were not on any active medication. Subsequently, resting ECGs were recorded, and echocardiographic examinations were performed, revealing that all participants had normal baseline ECG and echocardiography findings.

This study received approval from the Ethical Committee of Research at the George Emil Palade University of Medicine, Pharmacy, Science, and Technology of Targu Mures, Romania (CEC 129/2018).

### 2.2. ECG Recordings and Treadmill Exercise Testing

In the 6 h prior to the stress test, participants were instructed to avoid consuming alcohol, coffee, or any supplements known to have a stimulating effect.

Each participant underwent a resting standard 12-lead ECG recording, conducted in a supine position, after 10 min of rest. This was followed by a treadmill exercise test using the Bruce protocol, the details of which are shown in Table 1. The exercise testing was conducted until at least 85% of the age-predicted maximum heart rate was achieved.

Data recording was performed with a BTL-08 SD3 ECG device, which was connected to a desktop computer running the BTL CardioPoint software, version 2.27.24476.0. The sampling frequency of the ECG device was 2000 Hz, with a digital resolution of 3.9 μV and an A/D conversion of 13 bits.

### 2.3. Analysis of ECG Parameters

All ECG measurements were performed at baseline, at peak exercise, and at the end of the first minute of the recovery period. Leads II, III, V2, and V5 were selected for measurements because they provided the most suitable signals for analysis. 

The RR interval, P-wave duration, PQ (R) interval, QRS duration, QT interval, and T-peak to T-end time were measured. The repolarization-related intervals (QT, T-peak to T-end) were also corrected for heart rate using the Bazett formula. In addition, we evaluated the presence of atrial and ventricular ectopic beats. 

In the ERP+ group, characteristics of the J waves were also assessed, including the amplitude, localization, and morphology of the J wave and the slope of the ST segment.

Measurements were conducted manually, following the current recommendations [18]. During the measurements, the region of interest on the tracing was magnified, and a digital caliper was utilized. The mean value obtained from 5 consecutive beats was calculated for each parameter at baseline, peak exercise, and during the recovery period.

### 2.4. Statistical Analysis

Descriptive and inferential statistical analyses were performed. Continuous variables were expressed as means ± standard deviations if they were normally distributed, as verified by the Shapiro–Wilk test, or as medians with interquartile ranges (1st quartile–3rd quartile) for non-normally distributed data. 

Comparisons between groups (ERP+ vs. ERP−) were performed using Student’s *t*-test for normally distributed variables and the Mann–Whitney U-test for non-normally distributed variables. Categorical variables were presented as percentages, and between-group differences were assessed using the Chi-square test or Fisher’s exact test when required. A *p*-value of less than 0.05 was considered statistically significant. 

Data analysis was performed using Microsoft Excel, version 2402 (Microsoft Corporation, Redmond, WA, USA) and IBM SPSS Statistics, version 25 (IBM, Armonk, NY, USA).

## 3. Results

### 3.1. Clinical Characteristics

In the comparative analysis between the ERP+ and ERP− groups regarding their clinical characteristics, there were no significant differences in terms of the anamnestic profile (Table 2) or regarding age and anthropometric characteristics (Table 3).

During the evaluation of vital signs, the mean resting systolic and diastolic blood pressure levels were near the optimal blood pressure range for both groups, with no significant differences detected. In the ERP+ group, blood pressure exceeded the normal thresholds for two individuals, while in the ERP− group, this was observed in one individual. Except for one case, heart rates were within the normal range. However, when comparing the groups, a significant difference was observed, with the ERP+ group showing a higher average heart rate (Table 3).

### 3.2. Echocardiographic Characteristics

The echocardiographic examination revealed that all participants had normal baseline echocardiography findings, with no significant differences between the ERP+ and ERP− groups (Table 4).

### 3.3. Treadmill Exercise Testing

In the evaluation of exercise-induced physiological responses (Table 5), both the ERP+ and ERP− groups achieved high MET values. A significant majority (87% of the ERP+ group and 94% of the ERP− group) surpassed the threshold of 12 METs, which revealed an excellent exercise capacity. 

The mean peak heart rate during the treadmill test was 186 bpm for both groups. Additionally, the pressor response, marked by a systolic blood pressure at peak effort exceeding 160 mmHg and a modest rise in diastolic blood pressure at peak, was in the normal range.

### 3.4. Exercise-Induced ECG Changes

Figure 2 shows a representative example of the ECG modifications observed in individuals with and without ERP during exercise. The baseline values of ECG variables, along with their alterations throughout the exercise and recovery phase, are detailed in Table 6. These variations are further illustrated graphically in Figure 3, Figure 4, Figure 5, Figure 6 and Figure 7.

#### 3.4.1. P-Wave Duration

At baseline, the P-wave duration showed no significant difference between the ERP+ group and the control subjects. 

During exercise, the P wave shortened in both groups; nevertheless, this reduction was more pronounced in the control subjects. In the recovery phase, an increase in the P-wave duration was observed in both groups, and the significant difference between them began to decrease, with the *p*-value approaching the significance threshold (*p* = 0.043) one minute after the recovery phase.

#### 3.4.2. PR Interval

At baseline, there was no difference in the PR interval between the ERP+ and the ERP− groups. 

During peak exercise, both groups showed a shortening of the PR interval, with the interval being shorter in the control subjects; however, this difference did not reach statistical significance (*p* = 0.076). In the recovery phase, the PR interval lengthened, and the *p*-value moved further away from the significance threshold.

#### 3.4.3. QRS Duration

The duration of the QRS complex was shorter in the ERP+ group, with the difference being significant at baseline, during peak exercise, and in the recovery phase. 

It was observed that the duration of the QRS complex decreased during peak exercise and approached baseline values during the recovery phase in both groups.

#### 3.4.4. QT Interval

At baseline, the QT interval differed among the groups, being shorter in the ERP+ group. During exercise, the QT interval decreased in both groups, with no significant difference between them. During recovery, the QT interval increased in both groups. 

Regarding QTc duration, there was no significant difference between the groups at baseline, peak exercise, and during the recovery phase. However, the corrected QT interval was longer at peak exercise and during the recovery period, although this difference was not statistically significant.

#### 3.4.5. T-Peak to T-End Interval

The T-peak to T-end interval was significantly shorter in the ERP+ group before exercise, but the significance level was barely exceeded (*p* = 0.047); this difference disappeared during peak exercise and the recovery phase. 

The T-peak to T-end interval, corrected with the Bazett formula, was similar between the groups both at baseline and during exercise and in the recovery phase.

#### 3.4.6. Atrial and Ventricular Premature Beats

In total, atrial extrasystoles were observed in two subjects in the baseline ECG, one in each group. No atrial extrasystoles were observed at peak exercise and in the recovery phase. 

Ventricular extrasystoles did not occur in either group at baseline, during peak exercise, or during the recovery phase.

#### 3.4.7. J-Wave Characteristics

In the ECGs of the 23 individuals diagnosed with ERP, the average J-wave peak was 1.72 ± 0.4 mm, in 6 cases exceeding 2 mm. As for the ERP type, more than two-thirds were classified as type 2, which is characterized by an infero/infero-lateral J-wave localization. Regarding the morphology of the J waves, the notch type was slightly frequent (13 vs. 10 cases), and for the ST-segment slope, the majority (60.8%) were horizontal or downward (Table 7).

Examining the presence of the J wave during the different stages of the stress test, it was observed that the J wave either disappeared or did not meet the Macfarlane criteria [17]. At the end of the first stage of the treadmill exercise, about 40% of cases met the ERP criteria, and at the end of the second stage, less than 10% did. Following this, the J wave disappeared and only reappeared in the recovery phase (Figure 8).

## 4. Discussion

In this study, our aim was to assess the ECG parameter modifications related to exercise, which could have predictive value for arrhythmic events in individuals with ERP. Given the higher prevalence of ERP among young males and in order to eliminate gender-related ECG variations, this study was limited to male participants. 

The ERP+ group was matched with a control group, herein referred to as the ERP− group, whose ECGs showed no ERP. The groups had similar anamnestic profiles and anthropometric parameters. They shared identical family and personal medical history, showed similar mental health parameters, and their behaviors and physical activities were also the same. No significant differences were observed in age, weight, height, and body surface area, and both groups had similar resting blood pressure. This congruence in anamnestic and clinical parameters indicates that these variables probably did not influence the interpretation of the differences observed in ECGs between the groups.

Parameters from the treadmill exercise test demonstrated strong cardiovascular fitness, which is expected in a healthy young population. Participants in both groups displayed similar chronotropic and pressor responses to high-intensity exercise. The heart rate and blood pressure were within the expected ranges for high-intensity exercise, indicating no significant deviations from standard physiological responses in young men [19]. Our findings indicate that the presence of ERP does not affect the chronotropic and pressor responses during exercise.

Heart rate significantly influences electrical stability through its impact on action potential membrane currents, intracellular Ca^2+^ dynamics, and cellular energy levels. It also reflects the autonomic balance that influences these factors. Studies demonstrate an independent association between increased heart rate and ventricular arrhythmogenesis [20,21]. While most studies report no difference in baseline heart rate between individuals with ERP and control groups [3,4,10], some findings suggest that individuals with ERP may achieve a lower maximum heart rate during exercise [22]. Our results show a slightly higher resting heart rate in individuals with ERP; however, this difference disappeared during peak exercise and in the recovery phase, indicating an adequate sympathetic response to exercise in these individuals.

We examined various intervals on the ECG that characterize the impulse conduction in the heart at baseline, during peak exercise, and in the post-exercise recovery period. There are limited data in the literature about the effect of exercise on ECG parameters among individuals with ERP.

Intra-atrial conduction is characterized by the duration of the P wave and partly by the PR interval. Our results indicate that atrial conduction is similar at baseline and accelerates during exercise in both groups, as was expected in this population [23]. Notably, atrial conduction during peak exercise was significantly slower in subjects with ERP, a difference that persisted at the beginning of the recovery phase, although it decreased and approached the significance threshold. This trend was also observed when evaluating the PR interval, although the difference between the two groups for this parameter did not reach significance. Slower intra-atrial conduction is a well-recognized precursor of atrial arrhythmogenesis [24].

Ventricular depolarization and activation are represented by the QRS complex. Prolonged QRS duration is associated with an increased risk of sudden cardiac death in the general population as well as in individuals with Brugada syndrome [25,26]. It is noteworthy that early repolarization syndrome shares several clinical similarities with Brugada syndrome, and these two pathologies are described in the recent consensus report as the two forms of J-wave syndrome [27]. Paradoxically, our results showed a significantly shorter QRS complex duration in individuals with ERP, both at rest and during exercise. This finding suggests that ventricular depolarization is unlikely to contribute to arrhythmogenesis in individuals with ERP. However, it should be noted that in large-scale studies, QRS duration was identical in individuals with and without ERP [2,3,7].

The repolarization phase of the ventricular myocardium is well characterized by the QT interval and the T-peak–T-end interval. A prolonged QT interval increases the risk of malignant ventricular arrhythmias [16,25], and prolongation of the T-peak–T-end interval has been associated with arrhythmogenesis in J-wave syndromes [28,29]. Our results showed a shorter resting QT time in the ERP+ group, a difference that can be explained by the higher resting heart rate, as it disappeared when comparing the baseline corrected QT interval (QTc) and during exercise, when the heart rate of both groups reached similar values. The T-peak–T-end interval was slightly shorter in the ERP+ group at rest, but this value barely exceeded the significancy threshold. In addition, the T-peak–T-end interval was identical between the groups when corrected with the Bazett formula during the exercise phase.

In our study, we also investigated the effects of exercise on the J wave. It was observed that the J wave disappeared from the ECG during the treadmill test or did not meet the criteria required for ERP diagnosis. A trend was also observed between the degree of exercise and the disappearance of the J wave. Furthermore, the J wave reappeared in the post-exercise period. Similar results were reported by Nouraei et al., who investigated the effect of exercise on the ECG criteria for ERP in their study [30]. The disappearance of the J wave during exertion is probably explained by the increased sympathetic activity. It is known that high vagal tone is associated with the occurrence of ventricular fibrillation, and vagal tone is also increased in individuals with ERP [31,32,33]. Lowering vagal tone could therefore potentially reduce arrhythmogenesis in individuals with ERP.

The main limitation of the present study was the small number of cases, which could lead to a type I error in the statistical analysis. In addition, this study focused exclusively on a young male population. To validate these results, a larger sample and the inclusion of both genders are needed.

## 5. Conclusions

In healthy young males, the presence of an ERP has no impact on the chronotropic and blood pressure response during exercise. A slowed intra-atrial conduction was observed during exercise in individuals with ERP, which could serve as a predictive factor for atrial arrhythmogenesis. Furthermore, the QRS complex, QT interval, and T-peak–T-end interval were shorter in individuals with ERP in the baseline ECGs. During exercise and the recovery phase, the QRS complex remained shorter, but the differences in the repolarization parameters disappeared. This finding suggests that exercise-related changes in ventricular depolarization and repolarization are unlikely to contribute to arrhythmogenesis in individuals with ERP. It is noteworthy that the J wave disappears due to the increased sympathetic activity during exercise, a phenomenon with potential clinical significance.

## Figures and Tables

**Figure 1 diagnostics-14-00980-f001:**
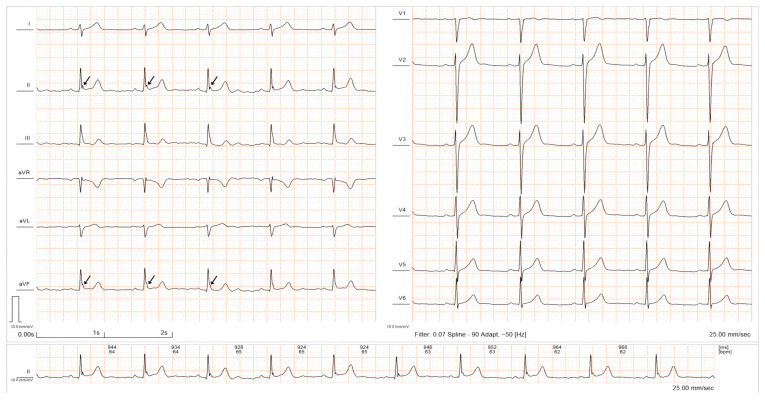
ECG recording showing ERP with a notching morphology in the inferior leads (from the authors’ personal collection).

**Figure 2 diagnostics-14-00980-f002:**
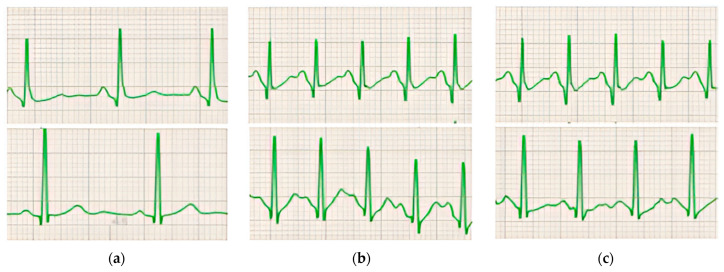
ECG examples obtained during exercise. Typical ECGs (lead II) recorded within 1 individual from each group (ERP+ in the upper row, ERP− in the lower row) at baseline (**a**), at peak exercise (**b**), and during recovery from exercise (**c**). The standard 25 mm/s and 10 mm/mV calibrations were used.

**Figure 3 diagnostics-14-00980-f003:**
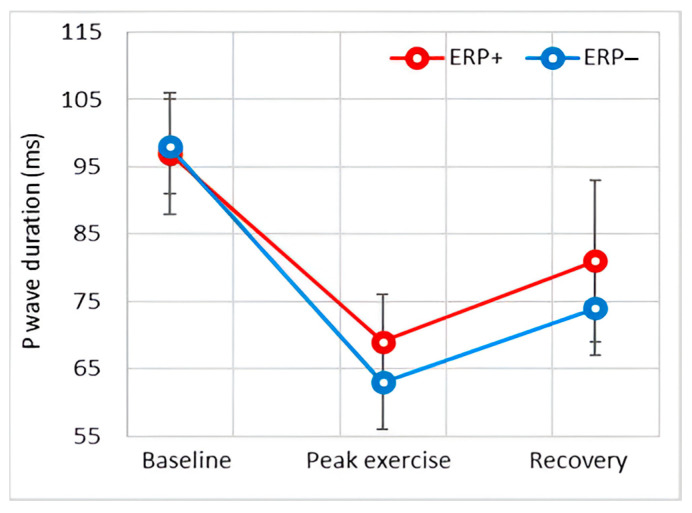
Effect of exercise on P-wave duration.

**Figure 4 diagnostics-14-00980-f004:**
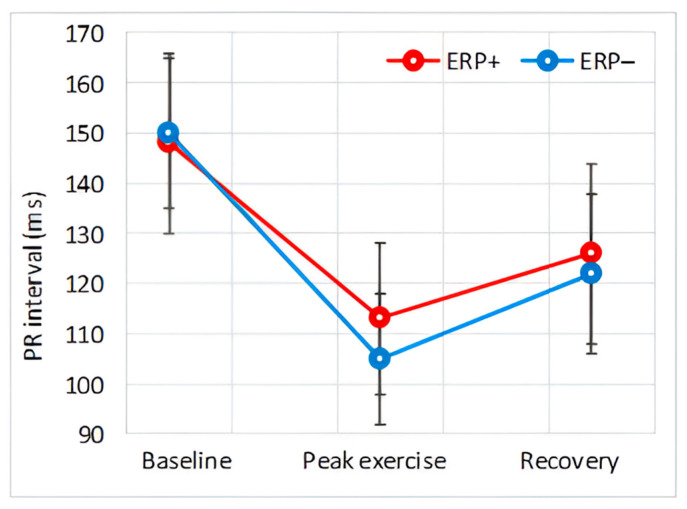
Effect of exercise on PR interval.

**Figure 5 diagnostics-14-00980-f005:**
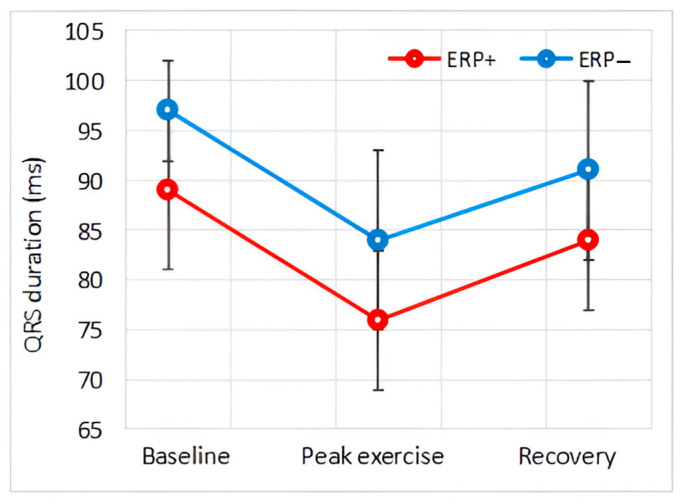
Effect of exercise on QRS duration.

**Figure 6 diagnostics-14-00980-f006:**
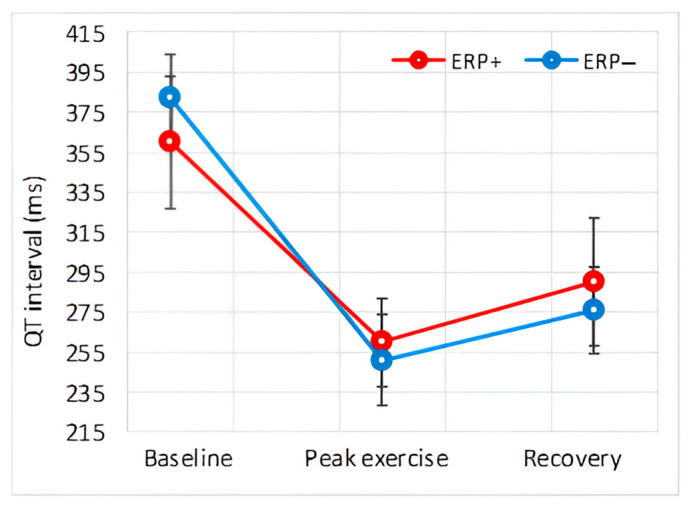
Effect of exercise on QT interval.

**Figure 7 diagnostics-14-00980-f007:**
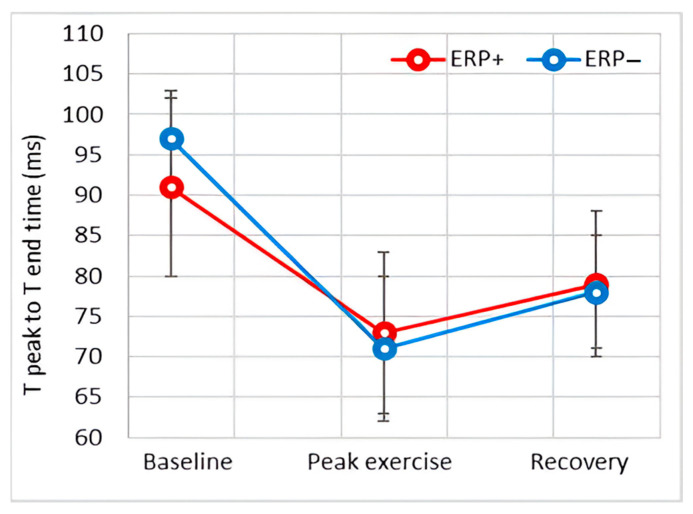
Effect of exercise on T-peak to T-end interval.

**Figure 8 diagnostics-14-00980-f008:**
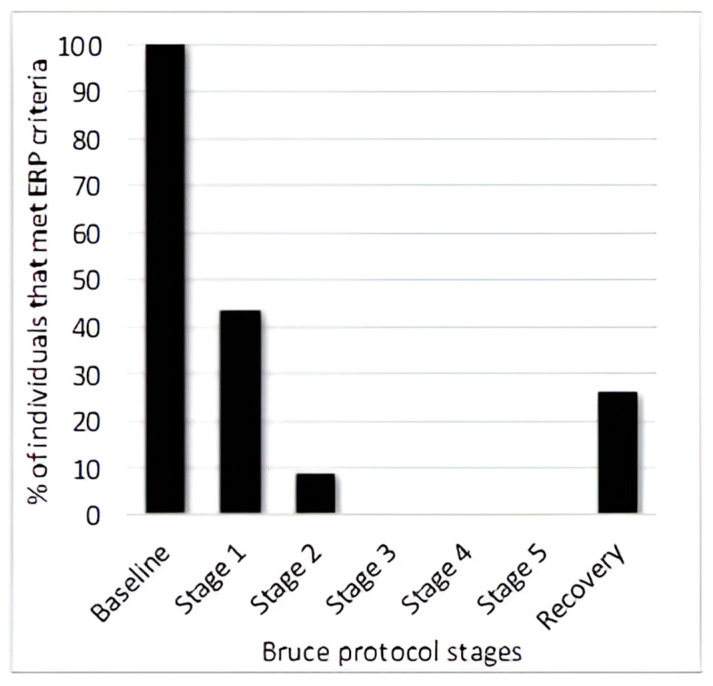
The percentage of participants meeting ERP criteria at various stages of exercise and recovery.

**Table 1 diagnostics-14-00980-t001:** The Bruce protocol used for graded treadmill exercise testing in this study.

Stage	Time (min)	Speed (km/h)	Slope (%)	METs
Before starting	1	0.8	0	1.4
Stage 1	3	2.7	10	4.6
Stage 2	3	4.0	12	7.0
Stage 3	3	5.4	14	10.1
Stage 4	3	6.7	16	13.4
Stage 5	3	8	18	17.2
Stage 6	3	8.8	20	20.3
Stage 7	3	9.6	22	23.7
Recovery	1	0.8	0	1.4

METs: Multiples of resting metabolic equivalent.

**Table 2 diagnostics-14-00980-t002:** Comprehensive anamnesis profile of individuals with and without ERP.

	ERP+	ERP−	*p*
**Family history**			
Sudden Cardiac Death, %	4.8	15.8	0.331
Arrhythmia, %	19.0	26.3	0.712
Myocardial infarction, %	4.8	21.1	0.172
Hypertension, %	47.6	68.4	0.184
**Personal history, after exertion**			
Chest pain, %	4.8	5.3	0.942
Dyspnea, %	33.3	36.8	0.816
Palpitations, %	38.1	36.8	0.935
Syncope, %	4.8	0	0.525
Dizziness, %	23.8	36.8	0.369
**Mental health**			
Stress level, scale 1–10	5 (3–5.5)	5 (3.5–5)	0.759
Anxiety, scale 1–10	2 (1.5–3.5)	2 (2–3)	0.988
Aggressiveness, scale 1–10	2 (1–3)	2 (1–4)	0.978
Mood, scale 1–10	9 (8–9)	8 (6.5–9)	0.427
Sleeping quality, scale 1–10	8 (6.5–8.5)	8 (7–9)	0.644
**Behaviors**			
Tobacco use, %	25.0	22.2	0.573
Alcohol consumption, %	90.0	66.7	0.117
Coffee intake, %	35.0	41.2	0.745
**Physical Activity**			
Sport activity, %	65.2	73.7	0.555
Weekly exercise, %	60.9	73.7	0.381

Data are presented as percentages or as medians with interquartile ranges (1st quartile–3rd quartile).

**Table 3 diagnostics-14-00980-t003:** Physiological measurements at the physical examination before the effort test.

	ERP+	ERP−	*p*
**Anthropometric measurements**			
Age, years	22.9 ± 1.6	22.1 ± 1.9	0.146
Weight, kg	78.4 ± 9.9	75.3 ± 10.4	0.328
Height, cm	178 ± 7	181 ± 7	0.275
Body surface area, m^2^	1.97 ± 0.15	1.93 ± 0.16	0.449
**Vital signs**			
Systolic blood pressure, mmHg	123 ± 12	122 ± 14	0.755
Diastolic blood pressure, mmHg	78 ± 6	78 ± 8	0.891
Heart rate, bpm	79 ± 14	69 ± 12	0.011

Data are presented as mean ± standard deviation.

**Table 4 diagnostics-14-00980-t004:** Echocardiographic measurements of the study population before the effort test.

	ERP+	ERP−	*p*
**Markers of the left ventricular structure**			
LVEDD, mm	45.8 ± 4.8	46.8 ± 3.7	0.459
LVESD, mm	27.5 ± 4.5	28.7 ± 3.3	0.314
IVS, mm	9.5 ± 1.1	9.5 ± 1.2	0.864
LVPWD, mm	9.2 ± 0.9	9.4 ± 1.0	0.638
LVM, g	147.3 ± 31.8	153.1 ± 29.5	0.555
**Markers of the left ventricular systolic and diastolic function**			
E/A ratio	1.6 ± 0.3	1.9 ± 0.4	0.068
DT, ms	208.3 ± 58.5	188.2 ± 59.5	0.290
E′, cm/s	14.2 ± 3.1	15.3 ± 1.9	0.198
E/E′ ratio	6.6 ± 2.2	5.8 ± 1.0	0.179
EF (Simpson), %	61.9 ± 5.8	60.3 ± 5.7	0.385
GLS	−21.1 ± 3.3	−20.7 ± 3.0	0.652
**Other echocardiographic parameters**			
LAD, mm	33.1 ± 5.3	32.3 ± 3.5	0.573
AoR, mm	21.3 ± 1.9	21.7 ± 1.8	0.479
RVD, mm	35.4 ± 4.6	33.5 ± 3.0	0.133
TAPSE, mm	25.1 ± 4	24.4 ± 4.5	0.616

Data are presented as mean ± standard deviation. LVEDD—left ventricular end-diastolic diameter; LVESD—left ventricular end-systolic diameter; IVS—interventricular septum; LVPWD—left ventricular posterior wall thickness; LVM—left ventricular mass; E—peak early diastolic transmitral flow velocity; A—peak late diastolic transmitral flow velocity; DT—deceleration time; E’—peak velocity of early diastolic mitral annular motion; EF—ejection fraction; GLS—global longitudinal strain; LAD—left atrial diameter (antero-posterior); AoR—aortic root dimension; RVD—right ventricular dimension (basal); TAPSE—tricuspid annular plane systolic excursion.

**Table 5 diagnostics-14-00980-t005:** Treadmill exercise testing parameters.

	ERP+	ERP−	*p*
METs	14.6 ± 2.7	16.5 ± 3.5	0.056
Duration of exercise, min	11.4 ± 2.3	12.2 ± 2.8	0.348
Heart rate at peak, bpm	186 ± 9	186 ± 8	0.977
Systolic blood pressure at peak, mmHg	170 ± 9	162 ± 13	0.180
Diastolic blood pressure at peak, mmHg	91 ± 12	83 ± 7	0.126

Data are presented as mean ± standard deviation. The MET values correspond to the level of exercise which was necessary to reach the target heart rate.

**Table 6 diagnostics-14-00980-t006:** ECG parameters at baseline, peak exercise, and during recovery from exercise.

	ERP+	ERP−	*p*
**RR interval, ms**			
Baseline	777 ± 157	899 ± 187	0.027
Peak exercise	321 ± 16	322 ± 14	0.972
Recovery	369 ± 58	374 ± 26	0.726
**P-wave duration, ms**			
Baseline	97 ± 7	98 ± 9	0.564
Peak exercise	69 ± 7	63 ± 7	0.014
Recovery	81 ± 7	74 ± 12	0.043
**PQ(PR) interval, ms**			
Baseline	148 ± 15	150 ± 18	0.678
Peak exercise	113 ± 13	105 ± 15	0.076
Recovery	126 ± 16	122 ± 18	0.579
**QRS duration, ms**			
Baseline	89 ± 5	97 ± 8	0.001
Peak exercise	76 ± 9	84 ± 7	0.009
Recovery	84 ± 9	91 ± 7	0.005
**QT interval, ms**			
Baseline	360 ± 22	382 ± 33	0.016
Peak exercise	260 ± 23	251 ± 22	0.107
Recovery	290 ± 22	276 ± 32	0.387
**QT interval corrected, ms**			
Baseline	412 ± 24	405 ± 24	0.387
Peak exercise	463 ± 38	443 ± 34	0.094
Recovery	484 ± 63	451 ± 46	0.071
**T-peak-T-end, ms**			
Baseline	91 ± 6	97 ± 11	0.047
Peak exercise	73 ± 9	71 ± 10	0.562
Recovery	79 ± 7	78 ± 9	0.590
**T-peak–T-end corrected, ms**			
Baseline	105 ± 14	103 ± 15	0.783
Peak exercise	129 ± 16	126 ± 19	0.543
Recovery	133 ± 20	128 ± 14	0.355
**Atrial premature beats**			
Baseline, %	5.3	4.3	0.706
Peak exercise, %	0	0	-
Recovery, %	0	0	-
**Ventricular premature beats**			
Baseline, %	0	0	-
Peak exercise, %	0	0	-
Recovery, %	0	0	-

Data are presented as means ± standard deviation or percentage. The corrected intervals were calculated using Bazett’s formula.

**Table 7 diagnostics-14-00980-t007:** The characteristics of the J wave prior to exercise.

Characteristic	ERP+ Group (*n* = 23)
J-wave peak, mm	1.72 ± 0.4
**ERP type**	
type 1, *n* (%)	7 (30.4)
type 2, *n* (%)	16 (69.6)
**J-wave morphology**	
notch, *n* (%)	13 (56.5)
slur, *n* (%)	10 (43.5)
**J-wave amplitude**	
≥2 mm, *n* (%)	6 (26.1)
<2 mm, *n* (%)	17 (73.9)
**ST-segment slope**	
horizontal/downward, *n* (%)	14 (60.8)
upward, *n* (%)	9 (39.2)

Data are presented as mean ± standard deviation (continuous variables) or as percentages (categorical variables).

## Data Availability

The dataset used for the current study is available from the corresponding author on reasonable request.

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
