# Peer review of "Exercise-Induced Electrocardiographic Changes in Healthy Young Males with Early Repolarization Pattern"

_diagnostics, 2024, doi:10.3390/diagnostics14100980_

Round 1

Reviewer 1 Report

Comments and Suggestions for Authors

Dear Editor,

Kocsis et al. provided an interesting manuscript entitled “Exercise‑induced electrocardiographic changes in healthy young males with early repolarization pattern” in which the authors performed an original study investigating ECG changes during exercise stress test in patients with early repolarization syndrome.

The manuscript is interesting with possible important clinical implications. The study included a group with early repolarization syndrome and a control group. Importantly, the two groups did not differ in terms of baseline characteristics. The study showed no differences in ECG parameters between groups during stress test, except for QRS duration which was increased in the ERP group.

The authors should add sampling frequency and sensitivity of the ECG device, and data regarding the echocardiographic measurements, particularly LVEF.

I recommend acceptance of the manuscript.

Reviewer 2 Report

Comments and Suggestions for Authors

The manuscript is devoted to the experimental research of ECG changes in young individuals with ERP during exercise in comparison with healthy individuals. The authors analyze ECG parameters including J wave during 7 stage physical load. They also pay attention to the statistical analysis of anamnesis profile and the results. The methods of analysis are standard. Such researches contribute to the base for the development of ECG analysis methods for diagnostics and prediction. Introduction part provides an extensive review of related works with the corresponding references. The references are relevant but 12 out of 33 are out of the last 10 years. The paper can be ascribed to the subject of the Diagnostics journal. This paper will be interesting for researchers in the field of ECG signal processing and contribute to ERP phenomenon research. On the whole the paper is well-prepared, well-structured and detailed. Some limitations of the study are also discussed (a small number of patients is the most significant one). The results can be used as a base for future researches with wider cohorts.

Some minor issues:

-         Line 35. Source [2] is dated 2008, looks not so recent.

-         Fig.1 and 2 – increase the resolution and mark the areas of interest.

-         Table 2 ‑ “Family history” – duplicate string.

-         Lines 251-253 – the paragraph looks strange, it is very short. It does not continue the previous one and does not start the next one. What is the sense of this isolated portion of information? It is better to move it one paragraph further.

-         Conclusion should be more detailed and correlated with the objectives of the study (the objective was formulated shortly).
